# Environmental and Endogenous Acids Can Trigger Allergic-Type Airway Reactions

**DOI:** 10.3390/ijerph17134688

**Published:** 2020-06-29

**Authors:** Giuliano Molinari, Laura Molinari, Elsa Nervo

**Affiliations:** 1Studio Tecnico Ing. Laura Molinari, Environmental Health and Safety Via Quarto Ponte 17, 37138 Verona, Italy; ing.laura.molinari@gmail.com; 2Elsa Nervo, Società Chimica Italiana, 00198 Rome, Italy; elsanervo@italway.it

**Keywords:** atmospheric acidity, air pollution, allergic reactions, mechanisms of allergy, allergic rhinitis, asthma, chronic, allergic multimorbidity, nonallergic, pseudo-allergic

## Abstract

Inflammatory allergic and nonallergic respiratory disorders are spreading worldwide and often coexist. The root cause is not clear. This review demonstrates that, from a biochemical point of view, it is ascribable to protons (H^+^) released into cells by exogenous and endogenous acids. The hypothesis of acids as the common cause stems from two considerations: (a) it has long been known that exogenous acids present in air pollutants can induce the irritation of epithelial surfaces, particularly the airways, inflammation, and bronchospasm; (b) according to recent articles, endogenous acids, generated in cells by phospholipases, play a key role in the biochemical mechanisms of initiation and progression of allergic-type reactions. Therefore, the intracellular acidification and consequent Ca^2+^ increase, induced by protons generated by either acid pollutants or endogenous phospholipases, may constitute the basic mechanism of the multimorbidity of these disorders, and environmental acidity may contribute to their spread.

## 1. Introduction

Inflammation and hypersensitivity of the airways and epidermis, whether allergic or nonallergic, acute or chronic, are pandemic illnesses and epidemiological studies show that they are growing faster in developing countries [1]. The increase has been attributed to several factors, both genetic [2] and environmental [3,4,5]; this work focuses on the latter.

Environmental factors that can affect the aetiology of these diseases, such as lifestyle, climate change, and air contaminants, have long been the subject of study and debate the world over [3,4,5,6,7,8,9]. The World Health Organization (WHO) has provided recommendations on how to reduce air pollution produced by household activities, one of these being to properly ventilate the home [9]. This is useful in rural areas but not in cities or industrial areas, where the outside air is often more polluted than the air indoors. Consequently, today, inflammatory allergic and nonallergic (also known as pseudoallergic) diseases are more widespread in urban than rural areas [10]. Authoritative research confirms that the higher prevalence in urban areas correlates with some outdoor air pollutants [3,10,11,12,13]. Immunological effects can be observed in both the upper and lower respiratory tract after exposure to diesel exhaust, and the short-term exposure to traffic-related nitrogen dioxide (NO_2_), an acidic gas, has a direct effect on respiratory morbidity [13]. Furthermore, a relationship between allergic diseases, Black Carbon (BC) and Particulate Matter (PM_2.5_) [14], and between air pollutants and allergic infant sensitization has been demonstrated [14]. The MeDALL (Mechanisms of the Development of Allergy) European study confirmed the relevance of environmental exposure [15,16]. Wherever possible, prevention by allergen avoidance remains the first measure [17]. Recent research has provided new data and technologies for therapeutic improvements [18]. However, further studies are needed to discover the molecular determinants and to clarify the basic onset mechanisms of allergic and nonallergic diseases [17,19].

In addition to the relevance of environmental exposures, the MeDALL study highlighted that air pollution not only correlates with bronchitis, rhinitis, asthma, and even eczema [10,11,12,13,14], but these diseases often co-exist and share causal mechanisms [15,16,20,21].

While the mechanism of allergic response has been extensively studied and remains mainly an IgE/FcεRI-based individual hypersensitivity reaction to specific allergens [1,6], there is no fully convincing biochemical explanation of the nonallergic response and the relationship between increasing allergic and nonallergic hypersensitivities, their multimorbidity, and air pollutants. IgE sensitization can no longer be considered the dominant causal mechanism of multimorbidity of such diseases [15,16,20], because allergic symptoms exist even in the absence of positive IgE tests. For these non-IgE-associated diseases, it is necessary to hypothesize other mechanisms, which should be investigated [6,16,20]. Some studies proved that this is in part attributable to genetic predisposition.

Regarding the consequences of environmental pollution, many studies have analyzed the toxic effects induced by air pollutants, in particular oxidative [22] and nitrosative [3,11] stress, and the causal relationship with allergies. Studies of acid stress began in the 1980s [23,24], without investigating the correlation between extra- and intracellular acidity. Acids can cause stress because they lower the physiological pH by the release of protons (H^+^).

The aim of this review is to highlight the chemistry of atmospheric acid pollutants, their irritating effects on the airways, and the existence of a possibly shared causal, proton-based mechanism, induced by both exogenous and endogenous acids, for the onset and spread of allergic and nonallergic inflammatory reactions.

Scientific literature available online from 1970–April 2020 was taken into consideration. The main databases, such as Embase, Medline, PubMed Central, Scopus, Web of Science, were searched and the most cited and most recent papers were selected. We analyzed the data and critically evaluated the fundamental biochemical concepts concerning the topic under study and their possible consequences on a cellular level.

## 2. Results

### 2.1. Outdoor Acid Air Pollutants: Chemical and Toxicological Characteristics

Polluting atmospheric acids damage surface water, buildings, and living organisms, either by direct reactions or through acid rain. Epidemiological studies on acute respiratory effects show that fine particulate matter (PM_2.5_) and gaseous acid pollutants can have a major impact on the airways [8,11,25,26], because of their significant toxic potential. Given their small size, fine particles are able to penetrate deeply and reach the lower respiratory airways [13,27]. Furthermore, as a result of their low polarity and high liposolubility, the gases can spread quickly through biological membranes [13,28] and hence enter cells. Among these gases, NO and O_3_ are known to cause nitrosative and oxidative stress, respectively. Recent studies have drawn attention to the health impacts of PM [13,14,26,27,29], NO_2_ [3,11,13,14,25,30], and SO_2_ [31,32,33].

It is known that PM from anthropogenic sources, such as heating systems, industrial plants, and motor vehicles, is mainly acid, since PM is associated with the anthropogenic acid pollutants NO_2_ and SO_2_ [11,13]. In addition, NO_2_ and SO_2_ can react with water and oxygen to give the corresponding acids: nitric acid (HNO_3_), sulphurous acid (H_2_SO_3_), sulphuric acid (H_2_SO_4_), and their related acidic salts. The toxicity of acid compounds is mainly due to their ability to release protons (H^+^). Both HNO_3_ and H_2_SO_4_ are strong acids, important sources of protons, and therefore are fiercely corrosive. While, at normal temperature and pressure, NO_2_ and SO_2_ are gases, the corresponding acids HNO_3_, H_2_SO_3_, and H_2_SO_4_ are liquids, and easily soluble in water. Their acidic salts are water-soluble as well. Therefore, most air acidity is concentrated in the microscopic PM, suspended in the air itself, in the form of both moist solid particles and watery droplets, known as acid aerosols. Notably, for its smaller particle size and its larger specific surface area, PM_2.5_ is richer than PM_10_ in water and acids.

Around the year 1985, interest in the effects of acid aerosols increased as a result of the risk of high exposure levels in the US and Canada. Clinical studies were carried out to assess the toxicity of some atmospheric pollutants. The results showed that:(a)The bronchoconstrictor action of carbachol could be enhanced by acid sulphate aerosols [23,34], even though the sulphate is not itself toxic [34];(b)The biologically active portion of these compounds is H^+^ rather than sulphate and the potency is proportional to their acidity [34];(c)Titratable acidity appears to be a more important stimulus to bronchoconstriction than pH [35].

Consistently, it was shown that bronchoconstriction provoked by inhalation of sodium sulphite aerosols was caused by the released gaseous SO_2_, or by bisulphite, but not by sulphite [35]. Combined exposures to acidic aerosols and pollutant gases have synergic effects.

The abovementioned PM, HNO_3_, H_2_SO_3_, and H_2_SO_4_ are the strongest acid components of acid aerosols. In addition, some other weaker acids are present, including carbonic acid, nitrous acid, and hydrogen sulphide, which essentially contribute to titratable acidity. All acids can contribute to the effects of total air acidity by releasing H^+^s to different extents.

### 2.2. Biochemical Effects of Cellular Acidification in Epithelial Tissues

It has been shown that exogenous acids can cause irritation and the bronchoconstriction of the airways [23,24,34,35,36] in both asthmatic [24] and healthy subjects [36]. Moreover, they can stimulate both immune cells (mast cells [37], neutrophils [38,39,40], dendritic cells [41], eosinophils [42], Jurkat cells, and primary T cells [43,44,45]) and nonimmune cells (epithelial cells [46,47], fibroblasts [20], and smooth airway muscle cells [48,49,50]). It is reasonable to assume that the effects of limited exposure by healthy subjects are negligible, because air acidity can be entirely neutralized within a short time by the buffering capacity of airway surface liquid (ASL) [51]. On the contrary, major exposure, or for sensitive people even limited exposure, can overcome the ASL defense, giving rise to the transfer of H^+^ into cells as described below.

Regarding endogenous acids, it should be remembered that cells use intracellular enzymes such as phospholipase C (PLC) and messengers such as inositol 1,4,5-trisphosphate (IP_3_) to increase the free Ca^2+^ concentration in cytosol ((Ca^2+^)_c_). PLC and other phospholipases are powerful acidifying enzymes, because one H^+^ is released for each hydrolyzed ester bond [52,53,54]. The hydrolysis of phospholipid esters and the generation of endogenous acid molecules, such as arachidonic acid (AA), phosphatidic acid, and IP_3_, are at the base of the production of allergic mediators. It is known that external acidification can cause mobilization of the segregated Ca^2+^ from intracellular stores [38,39,46,47,53,55,56,57,58,59], because protons can readily replace Ca^2+^ in its ligand locations [53,60,61,62]. Moreover, it is known that the increase of [Ca^2+^]_c_ is involved in many physiological processes [63,64,65,66,67], but also in the triggering of pathological manifestations such as allergic responses [65], airway hyper-responsiveness (AHR) [66], and abnormal contraction and remodeling of airway smooth muscle (ASM) [67].

The two paragraphs below give a more detailed description of the biochemical mechanisms by which intra- and extracellular acidification take place and foster allergic reactions.

### 2.3. Intracellular H^+^: Intracellular Acidification May Be Caused by the Action of Phospholipases in the Cytosol or by Protons Entering the Cell through the Plasma Membrane

The cells responsible for triggering the allergic response, such as mast cells [6,65,68,69,70] and basophils [6,69,70,71], have numerous receptors sensitive to various agonists. These receptors can be classified historically as IgE-dependent and non-IgE-dependent receptors, based on their positive or negative response to immunoglobulins E (IgE) [72]. The best example of an IgE-dependent receptor is the high-affinity IgE receptor (FcεRI) [73,74]. Non-IgE-dependent receptors include the recently discovered G-Protein-Coupled Receptors (GPCR), which respond to less specific agonists [69,71,75,76,77,78]. Furthermore, in the GPCR group are Mas-related G-protein coupled receptors (MRGR) [72,76], protease-activated receptors (PAR) [79], and purinergic receptors [80]. Given the sheer number of GPCRs, many combinations with different agonists are possible. For example, GPR4, GPR65, GPR68, and GPR132 receptors may be activated by extracellular protons [48,49,55,81,82]. Alternatively, muscarinic agonists may stimulate the Gαq/11 subunits of the acetylcholine GPCRs [48]. So-called pseudoallergic agents also follow this route [83]. All individuals can respond via GPCR receptors to the stimulus of the agonist, but only sensitized individuals (the “truly” allergic) can respond via IgE/FcεRI.

Contact between the agonist and the receptor triggers a PLC/IP_3_-pathway-type complex chain reaction which, via the activation of numerous enzymes and the increase in the concentration of H^+^ and cytosolic Ca^2+^ (respectively (H^+^)_c_ and (Ca^2+^)_c_), culminates with degranulation, by the exocytotic secretion of allergic mediators and the onset of an acute allergic response. The responses of the various agonist/receptor couples may differ [69,77,84], but depend in each case on the concentration and affinity of the agonist [74], and the fundamental steps in the basic biochemical mechanism of allergic reactions do not vary (Figure 1):(a)The stimulation of the receptor, both of the FcεRI and GPCR types, activates phospholipase C (PLC) [85,86,87,88] and hence the hydrolysis of phosphatidylinositol 4,5-biphosphate (PIP_2_) on the inner wall of the plasma membrane, generating and releasing IP_3_, a protonated acid salt [62,89], in the cytosol;(b)Through dissociation, the IP_3_ releases H^+^ [62,89] and, via its IP_3_R receptor, induces cell calcium release and store depletion, increasing (Ca^2+^)_c_ [62,90];(c)The increase in (Ca^2+^)_c_ activates numerous calcium-dependent enzymes, including phospholipase A_2_ (PLA_2_), which produces arachidonic acid (AA) [91,92], which in turn dissociates releasing more H^+^ and inducing the release of more Ca^2+^ [56,58,93]; from the AA hundreds of derivatives (eicosanoids cascade) are formed, including leukotrienes (LTs) and prostaglandins (PGs) [94,95]. Both leukotrienes and prostaglandins are known to play a pivotal role in inflammatory and allergic reactions;(d)The store depletion stimulates the entry of more Ca^2+^ from the extracellular space (calcium influx) via the mechanism known as Store Operated Calcium Entry (SOCE), in which, from the surface of the Endoplasmic Reticulum (ER), Stromal Interaction Molecule1 (STIM1) activates the opening of ORAI1 and Transient Receptor Potential Cation Canonical (TRPC) [96,97,98,99] channels on the plasma membrane;(e)The calcium influx further stimulates PLA_2_ activity and fosters the maturing of the granules and subsequent degranulation and release [100,101,102,103] of mediators [94,104,105], including histamine, PGs, LTs, cytokine, tryptase, and chymase, which promote the acute phase of allergic inflammation. The cysteinyl LTs are thought to be responsible for the increase in the basal tone of the ASM and in bronchoconstriction in asthma [6,106].

In conclusion, as shown in Figure 1, the allergic and nonallergic responses differ only in the first step of agonist stimulation, which leads to PLC activation. The subsequent PLC/IP_3_ pathway is the same for both responses and is characterized by the generation of acids, such as IP_3_ and AA, and thus, H^+^ release by acid dissociation. Figure 1 shows two different steps in the intracellular generation of H^+^ through the action of phospholipase, the first dependent on the IP_3_ produced by PLC, the second on the AA produced by PLA_2_. A third step, not shown in Figure 1, can depend on the action of phosphatases which dephosphorilate the IP_3_ on IP_3_R [62]. Each of these three steps gives rise to a rapid transient increase in (H^+^)_c_. This increase, as a result of the protons derived from the IP_3_ and the AA, contributes to cell store depletion/calcium release and to the subsequent Ca^2+^ influx, via the activation of SOCE, with a consequent increase in (Ca^2+^)_c_ [39,47,56,57,58,59,62,107]. The rise in (H^+^)_c_ is transient because it is subject to feedback control and can be rapidly neutralized by the buffering capacity of cytosol and the calcium influx itself, which leads to the alkalization of the cytosol, because the extracellular pH outweighs the intracellular pH. The influx may also be induced by a mechanism other than SOCE and independently of the reserves, known as Store-Independent Calcium Entry (SICE), by direct activation, via STIM1 and ORAI, due to the AA or LTs [108].

In addition to being generated in the cytosol by the phospholipases as described above, H^+^ can penetrate the cell directly [28,38,46,109,110,111,112,113,114], passing through the epithelial barrier and plasma membrane, thanks to the acid loaders (Figure 2A). This is possible because the permeability of the epithelial barrier can vary as a result of the stimuli received from the cellular receptors [115], or the barrier itself may be destroyed [116,117]. Examples of acid loaders are the Cl^−^/HCO_3_^−^ exchangers of the SLC4 and SLC26 type, [117,118,119,120] and the Na^+^-HCO_3_^−^ cotransporter of the SLC4 type [118,120], which are chemically equivalent to a counterflux of H^+^ ions, the plasma membrane Ca^2+^ ATPase pump (PMCA), which exchanges H^+^ for Ca^2+^ [121], acid-sensing ion channels (ASICs) [53,82,122], ORAI [123], and some types of TRP channels [109,110,111,112]. Furthermore, H^+^ can be released into the cell after entry by passive transfer of reactive oxides coming from atmospheric pollution, such as CO_2_, NO_2_, and SO_2_. CO_2_ can react with water to release H^+^ much quicker than NO_2_ and SO_2_ due to the specific ubiquitous catalyst Carbonic Anhydrase (CA) [124]. Therefore, the CO_2_/CA system is possibly an excellent means of transport for H^+^, as some researchers believe [38,120,124]. Lastly, the extracellular excess of protons may enter the cell by diffusion [125].

### 2.4. Extracellular H^+^: The Acidification of the Surfaces of the Respiratory Airways May Be Due to Environmental Acid Pollutants or Endogenous Acids

The outer surface of the epithelia of respiratory pathways is kept moist at all times by ASL. ASL plays a key role in the defense of the airways from pathogens and contains some phagocytes and a number of proteic and peptidic antimicrobials for this purpose. For optimum antimicrobial activity, both in the nose and lungs, ASL pH should be maintained within slightly acidic physiological values (circa pH = 6.80) [24,126,127] and a lowering can be counterproductive [128]. Interestingly, a decrease in ASL pH after exposure to airborne traffic pollutants has been detected in asthmatic [128] and healthy subjects [129], albeit to different extents.

Historically, endogenous acidification of the airway surfaces has been suggested as a way to measure airway disease [126]. In 2000, Hunt et al. found that the pH of Exhaled Breath Condensate (EBC) was over two log orders lower in patients with acute asthma than in healthy subjects. Hence, they suggested a possible causal relationship between endogenous airway acidification and the airflow limitation observed in acute asthma [130]. Similarly, in more recent years, low pH values of ASL have been observed by other authors in asthma, allergic rhinitis, atopic dermatitis [131], and even in nonallergic inflammatory diseases, such as bronchiectasis and Chronic Obstructive Pulmonary Disease (COPD) [117].

Figure 2B shows the possible causes of the lowering of ASL pH depending on endogenous or exogenous acidity. Extracellular and ASL acidification may be caused in four different ways:(a)H^+^ derived from the physiological process of restoring prestimulus conditions, carried out by all cells through the expulsion of excess protons, generated by acidifying enzymes, to return to the steady state; cells can use acid extruders as exchangers and channels to transfer H^+^ externally; the Na^+^-H^+^ exchanger (NHE) in some cells is the major acid-extruder, also the Cystic fibrosis transmembrane conductance regulator (CFTR) plays an important role in the acidification of the ASL [117]; in addition, the excess protons in the cytosol may exit the cell via voltage-gated proton channels (Hv1), TRP channels, plasma membrane vacuolar V-type H^+^-ATPase [126,132,133,134,135,136], and diffusion [125];(b)The degranulation of phagocytes, such as macrophages and granulocyte neutrophils and eosinophils [69,135,137,138], produced as a defensive inflammatory action [24,126] in response to the stimulus. This acidifying action may be significant and long lasting, and is therefore the basis for chronic disease;(c)The degranulation of mast cells and basophils, caused by the stimulus, the basis of the acute allergic response [77,78,80,84,138], as described above in Figure 1. It is known that, like phagocytes, basophils and mast cells [138] can produce and secrete acids and phospholipolytic enzymes with the contents of their cytosolic granules and vesicles. Examples of secreted acids are lactic, hypochlorous, uric, phosphoric acid, and fatty acids. Examples of enzymes are the cytosolic and secretory phospholipases A2, which produce fatty acids such as AA through hydrolysis of cellular triglycerides and phospholipids [139]. Each of the secreted acids can contribute to the release of protons and thus act as new stimuli for cellular responses;(d)In addition to the endogenous acids described above in point a, band c, which are transferred by the cells to the ASL by means of expulsion, extrusion, and/or degranulation, the acidification of the ASL may be due to exogenous acids, and hence, possibly, to the presence and direct action of atmospheric acid pollutants.

All four processes of acidification described in points a–d, and in Figure 2B, can contribute separately or simultaneously to lowering ASL pH. It is known that protons are allosteric modulators and protein structure modifiers [140]. The harmful consequences of the lowering of ASL pH, caused by either exogenous or endogenous acids, can include an increase in mucus viscosity, a decrease of ciliary beat frequency, recruitment of proinflammatory mediators, oedema, leukocyte infiltration, AHR, tissue remodeling, and damage [24,40,50,136]. These consequences foster the origin and development of acute and chronic diseases of an allergic kind [24,37,40,66,72,130].

## 3. Discussion

### 3.1. Difficulties to Overcome

It is difficult to demonstrate if a common causal mechanism for the onset and increasing spread of inflammatory allergic and nonallergic diseases exists and how it works, but it is very important. To our knowledge, a great number of interesting publications are available in the literature on this theme, but none specifically takes into consideration inflammatory nonallergic manifestations from a biochemical point of view. The association of airway inflammation, bronchoconstriction, and/or asthma with acids [23,24,33,34,35,36,37,40,130], and more specifically, of allergic responses with some particular acids, such as sulphurous [33,35], and AA [6,58,141], has long been known. The association of environmental acids with allergic sensitization [37,142] and the hypothesis that these diseases might share a common mechanism [15,16,20] have been considered more recently. Some critical issues emerge from reading the existing toxicological studies. Important criticalities arise, above all, from the features of the proton (H^+^ ion) (small, very mobile and fast, able to interact with many molecular entities). These properties, which make it an ideal activation factor, are at the same time an obstacle to detection by normal instruments. In addition, the interdependence of the concentrations of H^+^ and Ca^2+^ suggests that the latter varies rapidly and in parallel with the former. This depends on the intrinsic chemical properties of the two ions [62]. Even in simple aqueous solutions without biological structures, it can be observed that the addition of acids quickly solubilizes the bound calcium and therefore produces an increase in (Ca^2+^), whilst the addition of alkali causes it to deposit and therefore reduces (Ca^2+^).

Other criticalities arise from the difficulty in isolating a single pollutant, and in the case of PM, its nonspecific composition. There are a number of variables at play, some of which are hard to investigate. Studies often include different cells and different experimental conditions in terms of method and duration. It is therefore very difficult to evaluate and compare data and conclusions. This should be carefully considered in experimental studies. However, modern instruments and techniques can help. In particular, biosensors, which allow one to study subcellular H^+^ and Ca^2+^ dynamics simultaneously, in combination with electron cryomicroscopy and X-ray crystallography should give interesting results.

One question arises spontaneously: “If acids play an important role in asthma and allergies, how are the pathologic responses to basic compounds to be explained?” As recently pointed out [83], most so-called pseudoallergic compounds are basic. The answer, given by the same author, is that pseudoallergic compounds activate G proteins, directly or through GCPRs [83]. Accordingly, the subsequent steps follow the abovementioned PLC/IP_3_ pathway, involving acid generation and thus release of H^+^.

Some other studies [123,143,144] showed that in vivo cellular alkalinization causes a substantial increase in (Ca^2+^)_c_.

This is not in contradiction with what is reported here (Section 2.3, the intracellular H^+^ paragraph), because different events are involved. In their article, Yu et al. [123] describe studies of the regulatory not the activation mechanisms, and some authors think the increase in (Ca^2+^)_c_ caused by the alkalinization may be due to influx [143] or to the inhibition of Ca^2+^ATPase and influx [144]. Influx causes an increase in intracellular calcium, since [Ca^2+^] is normally much higher outside than inside the cell. In Section 2.3, the intracellular H^+^ paragraph, we describe the rapid and transient increase in [Ca^2+^]_c_, caused by the activation of phospholipase or the entry of extracellular H^+^. Both of these events occur before influx.

### 3.2. Possible Deductions

This review on the acidification/increase of (H^+^) in external cells and epithelia highlights that the acidity of external epithelia can have both exogenous (environmental acidity) and endogenous (phospholipase activation) origins (Figure 2B), and therefore, the cellular calcium homeostasis can be altered from both outside and inside.

By the release of protons in various ways via acid loaders and acid extruders through the plasma membrane, the acids increase (Ca^2+^)_c_ and activate immune cells, inducing the inflammation of the airways and bronchospasm. The parallelism and interdependence of the concentrations of H^+^ and Ca^2+^, and the well-known ability of H^+^ to easily replace Ca^2+^ in its binding sites are the basic facts that suggest that the various means of intracellular acidification, of both exogenous and endogenous origin, have a common mechanism, with H^+^ acting as a stimulus for the increase in (Ca^2+^)_c_.

Many enzymes at the basis of the allergic and nonallergic inflammatory response have catalytic activity, strictly dependent on pH and/or Ca^2+^ as cofactors. The PLC and PLA_2_ themselves are Ca-dependent. Therefore, intracellular acidification, of both exogenous and endogenous origin, may induce acute inflammatory reactions and hypersensitivity through the activation of specific enzymes and the modulation of their action.

Furthermore, exogenous and/or endogenous acidification may favor the lowering of the ASL pH and the reiteration of the acid stimulus, triggering the recruitment of proinflammatory mediators and chronic disease. To sum up, it is possible that:(a)Environmental acidity increases the sensitivity of epithelial surfaces and promotes AHR;(b)Exogenous and endogenous acids contribute to both the decrease in ASL pH and the increase in ASM basal tone, thus favoring bronchoconstriction;(c)The excess of temporary intracellular acidification is at the origin of acute manifestations of an allergic kind;(d)Recurrent or continuous acidification is the biochemical basis of airway inflammation, hyper-responsiveness, tissue remodeling, and chronicity.

Accordingly, the impairment of H^+^/Ca^2+^ homeostasis and particularly their abnormally high concentrations can constitute a powerful biochemical basis for the onset, continuation, and multimorbidity of disorders, such as inflammatory allergic and nonallergic acute and chronic reactions. The entry/exit pathways for the protons, as described above, are based on physiological activation mechanisms and therefore could be carried out either in healthy or sensitized subjects. The variety of possible paths to increase and control intracellular H^+^ and its numerous interactions in the human organism require biomedical studies to explain the diversity of responses and existing situations.

A relation between oxidative or nitrosative stress and acid stress was also put forward [24]. Notoriously, protons can readily produce modifications in the conformation of proteic molecules [140]. Moreover, environmental pollutants have been associated with some asthma phenotypes through the mediation of IL-13 and DNA methylation [2]. DNA methylation is favored by heavy metals, which in turn are made available by acid mobilization. Certain metal constituents of PM_2.5_ were associated with circulating biomarkers of endothelial function [145]. Therefore, environmental acids might play a role in genetic/environmental interactions, by inducing epigenetic modifications with consequent allergic sensitization.

## 4. Conclusions

Acid pollutants can have toxic, cumulative effects on human epithelia via the release of protons. Protons can affect cellular homeostasis from both outside and inside. Therefore, it can be assumed that intracellular acidification, and the consequent increase in Ca^2+^ concentration induced by protons generated either by acid pollutants or endogenous phospholipases, may be at the basis of the shared causal mechanism of acid stress and multimorbidity of respiratory and hypersensitivity reactions. Moreover, acid environmental pollutants can contribute to the development and growing spread of inflammatory allergic and nonallergic reactions worldwide.

Further studies are required to clarify the specificity and the activation pathways of G proteins in general, and in relation to protons, considering the very high number of GPCRs discovered in recent years. Similarly, further studies are required into the ability of ion channels to transfer H^+^ into cells, together with an investigation of the permeability of plasma membranes to gaseous pollutants, such as NO, NO_2_, SO_2_, and particularly to CO_2_, because CO_2_ may have considerable influence on intracellular pH as well as on titratable acidity. Studies of acidity are often limited to measuring only pH, but the measurement of both pH and titratable acidity is indicated for better evaluation purposes.

Identifying and understanding the mechanisms of feedback and control of the different processes of cytosolic acidification, either of internal or external origin, temporary or lasting, and their consequences represents a major challenge for future research.

Reducing air acidity may be an important aim to limit the spread of the disorders taken into consideration in the present study, and to improve the health, especially in children and in frail subjects, of those more exposed to the risk of diseases. We believe our review calls attention to the fundamental importance of H^+^/Ca^2+^ interdependence and hope it contributes to further studies into allergic reactions and the identification of the molecular causes of these disorders.

## Figures and Tables

**Figure 1 ijerph-17-04688-f001:**
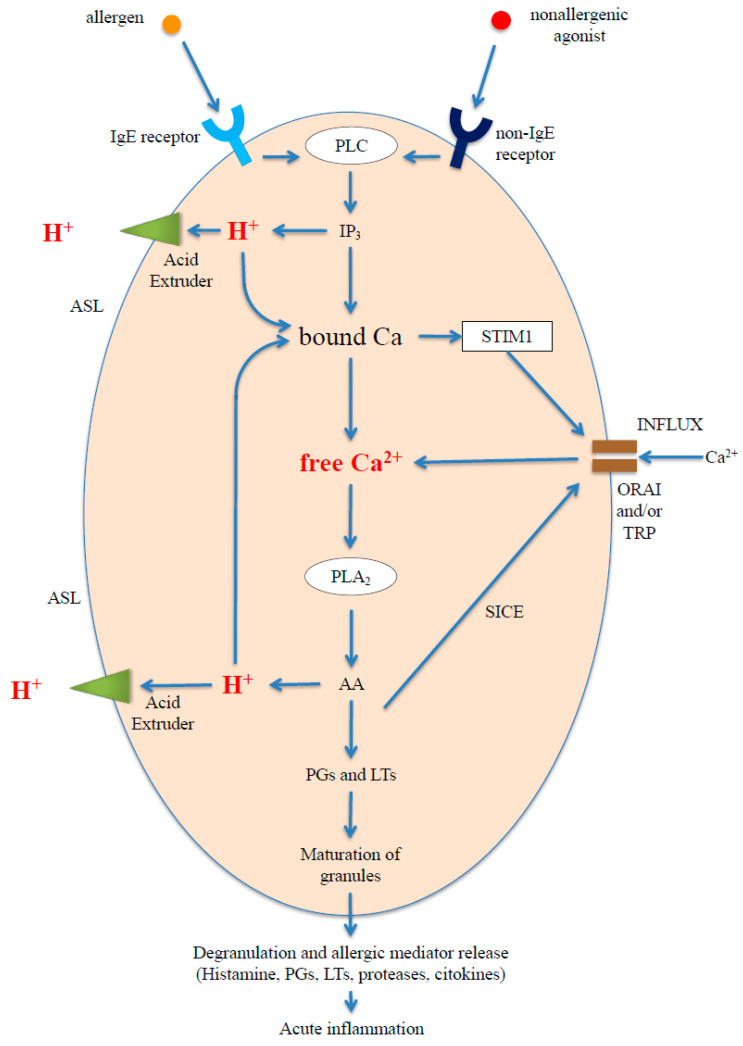
Basic biochemical mechanism of allergic-type response; PLC—Phospholipase C; PLA_2_—Phospholipase A_2_; AA—Arachidonic Acid; SICE—Store-Independent Calcium Entry; PGs—Prostaglandins; LTs—Leukotrienes; ASL—Airway Surface Liquid.

**Figure 2 ijerph-17-04688-f002:**
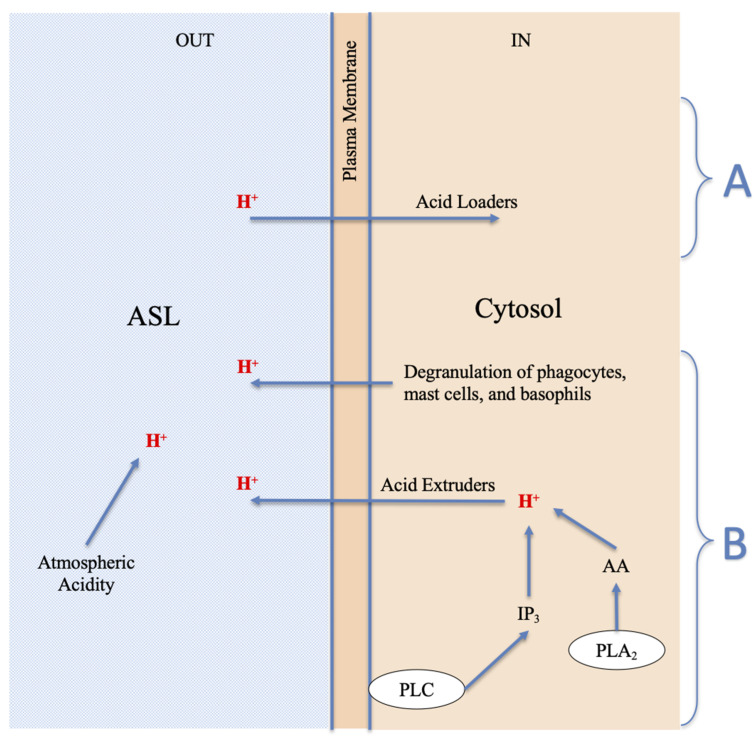
Biochemical routes for variation in pH of the Airway Surface Liquid (ASL). (**A**) Acid loaders: Cl^−^/HCO_3_^−^ exchanger; Na^+^- HCO_3_^−^ cotransporter; PMCA pump; TRP, ORAI and ASIC channels; CO_2_/Carbonic Anhydrase system, etc. (**B**) Acid extruders: NHE exchanger; Hv1, TRP, and CFTR channels; ATPase pumps, etc. PLA_2_—Phospholipase A_2_; PLC—Phospholipase C; IP3—Inositol 1,4,5-trisphosphate; AA—Arachidonic Acid.

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
