# Peer review of "Environmental and Endogenous Acids Can Trigger Allergic-Type Airway Reactions"

_ijerph, 2020, doi:10.3390/ijerph17134688_

Round 1
Reviewer 1 Report
The study looks at air acid pollutants and the irritating effects they have on the airways and assess the possible existence of a causal, proton-based mechanism which is induced by both exogenous and endogenous acids and affects allergic and nonallergic inflammatory reactions. The authors do a good job in explaining the biochemical mechanism of intra and extra cellular acidification causing allergic and non allergic reaction. Pathways are relatively explained well although somewhat complicated. I found some difficulties understanding the section in discussion (line 291 and after) regarding contradictions regarding the review of material showing links between acidification and increase in Calcium inside cells. Authors report on studies that show increase in intracellular Calcium with alkalization inside the cell but do not fully explain how this is not contraindicating their results.
Would emphasize more that this is a review throughout the manuscript.
Author Response
Response to Reviewer 1 Comments
We are grateful to the Reviewer for his attention and valuable comments.
Point 1: I found some difficulties understanding the section in discussion (line 291 and after)
Response 1: Some parts may be difficult to understand, because the formatting of the original submission has changed two numbered list. We propose:
Line 230, at the beginning, insert “ a) “
Line 238, at the beginning, replace “ a) ” with “ b) “
Line 242, at the beginning, replace “ b) ” with “ c) “
Line 251, at the beginning, replace “ c) ” with “ d) “; replace “ points 1, 2 and 3 “ with “ points a, b and c “
Line 255, replace “ points 1-4 “ with “ points a-d “
Line 291, delete “In contrast to new answers suggested in points A-D,“; replace “one“ with ”One“
Point 2: Authors report on studies that show increase in intracellular Calcium with alkalization inside the cell but do not fully explain how this is not contraindicating their results.
Response 2: We propose:
Line 302, replace “or the inhibition of Ca2+ATPase [144].” with “or to the inhibition of Ca2+ATPase and influx [144]. Influx causes an increase of intracellular Calcium, since [Ca2+] is normally much higher outside than inside the cell.”
Line 304, replace “extracellular H+, prior to the influx.” with “extracellular H+. Both of these events occur before the influx.”
Point 3: Would emphasize more that this is a review throughout the manuscript.
Response 3: We propose:
Line 11, replace “paper” with “review”
Line 28, replace “work” with “review”
Line 63: replace “work” with “review”
Line 306: replace “research” with “review”
Line 368: replace “research” with “review”
Reviewer 2 Report
The manuscript submitted for review is a comprehensive discussion of the problem of the effect pollution on the airways.
In my opinion, the review shuold be published, and many scientists will find answers to many questions, and the article will be helpful for further research. I have no objections and suggest publication.
Author Response
Response to Reviewer 2 Comments
Many thanks to the Reviewer 2 for his attention and his flattering comments.
Reviewer 3 Report
Dear authors,
This was an interesting manuscript, which can contribute to our knowledge of potential mechanisms of toxicity of air pollutants. However I do have a few points that you might want to review and consider:
1- In line 37, you said human nose, it is preferred if you use the terminology upper and lower respiratory tract or nasal region and lower airways.
2- In line 63, it should be chemistry
3- In line 67, you should state the methodology for the selection of the articles more that just the years. For example including the searched databases, keywords and criteria for inclusion/exclusion.
4- In line 75, it should be "given their small size, fine particles...." To this extent, throughout the manuscript sometimes the way the sentences are stated it's hard to know whether you are talking about the particles, the gases, or particle-absorbed gases.
5- In line 79 and others, it should be PM not PMs
6- Lines 81 to 90, this paragraph needs references, not all PM is acidic, it depends on the composition and source of the particles. If the whole basis of the articles is around exogenous acidic pollutants, this paragraph needs to be stronger in particular for PM.
7- Lines 106-107, although environmental legislation does not regulate acidity, it does regulate NO2, SO4, and PM. Also there are international regulations to lower sulphate aerosols through the use of low-sulphur fuel. So the statement it's a little bit misleading and really there is no way to regulate acidity per se, one can regulate chemical species and specific pollutants, and those contributing to acid rain are regulated.
8- Line 221, demonstrated is a strong word for a study in which only 40 subjects participated. Please change to found or observed.
9- Lastly, to strengthen the review please add a section of in-vitro, in-vivo and human studies in which the pH of the pollutants were studied and its relation to the potential mechanism you are highlighting in this manuscript.
Author Response
Response to Reviewer 3 Comments
We are grateful to the Reviewer 3 for his attention and valuable comments.
Point 1: In line 37, you said human nose, it is preferred if you use the terminology upper and lower respiratory tract or nasal region and lower airways.
Response 1: Lines 37, we have replaced “the human nose and the lower airways” with “the human upper and lower respiratory tract”
Point 2: In line 63, it should be chemistry
Response 2: Line 63, we have replaced “chemism” with “chemistry”
Point 3: In line 67, you should state the methodology for the selection of the articles more that just the years. For example including the searched databases, keywords and criteria for inclusion/exclusion.
Response 3: Line 67, after “consideration. ”, we propose to insert: “The main databases, such as Embase, Medline, PubMed Central, Scopus, Web of Science have been searched and the most cited and most recent papers have been selected.”
Point 4: In line 75, it should be "given their small size, fine particles...." To this extent, throughout the manuscript sometimes the way the sentences are stated it's hard to know whether you are talking about the particles, the gases, or particle-absorbed gases.
Response 4: Line 75, we have replaced “given their small size, they are” with “given their small size, fine particles are”
Yes, sometimes, when the specification seemed too long or complex, we were deliberately generic.
Point 5: In line 79 and others, it should be PM not Pms
Response 5: Lines 79, 87, 103, and 284 we have replaced “PMs” with “PM”
Line 284, we have replaced “their” with “its”
Point 6: Lines 81 to 90, this paragraph needs references, not all PM is acidic, it depends on the composition and source of the particles. If the whole basis of the articles is around exogenous acidic pollutants, this paragraph needs to be stronger in particular for PM.
Response 6: Line 81, we propose the replacement of “It is known that PMs are acidic.” with “It is known that PM from anthropogenic source such as heating systems, industrial plants and motor vehicles is mainly acid, since PM is associated with the anthopogenic acid pollutants NO2 and SO2 [11, 13].”
Point 7: Lines 106-107, although environmental legislation does not regulate acidity, it does regulate NO2, SO4, and PM. Also there are international regulations to lower sulphate aerosols through the use of low-sulphur fuel. So the statement it's a little bit misleading and really there is no way to regulate acidity per se, one can regulate chemical species and specific pollutants, and those contributing to acid rain are regulated.
Response 7: Lines 106-109, we propose to delete four lines, from “Current” to “level.”
Point 8: Line 221, demonstrated is a strong word for a study in which only 40 subjects participated. Please change to found or observed.
Response 8: Line 221, we have replaced “demonstrated” with “found”
Point 9: Lastly, to strengthen the review please add a section of in-vitro, in-vivo and human studies in which the pH of the pollutants were studied and its relation to the potential mechanism you are highlighting in this manuscript.
Response 9: We regret that we are unable to respond to this invitation at the moment. We fully agree that adding in-vitro, in-vivo and human studies related to our mechanism would strengthen the review. We looked for such studies in the literature; we haven't found any. We hope our review will promote them.